# A Disila[2]ferrocenophane with a Bridging 9,9′-Bi-9*H*-9-Silafluorene Moiety

**DOI:** 10.3390/molecules30061361

**Published:** 2025-03-18

**Authors:** Shinnosuke Usuba, Shogo Morisako, Koichiro Masada, Koh Sugamata, Takahiro Sasamori

**Affiliations:** 1Graduate School of Science and Technology, University of Tsukuba, 1-1-1 Tennoudai, Tsukuba 305-8571, Ibaraki, Japan; usuba@dmb.chem.tsukuba.ac.jp; 2Sagami Chemical Research Institute, Hayakawa 2743-1, Ayase 252-1193, Kanagawa, Japan; s-morisako@sagami.or.jp; 3Department of Chemistry, Institute of Pure and Applied Sciences, University of Tsukuba, 1-1-1 Tennoudai, Tsukuba 305-8571, Ibaraki, Japan; masada_k@chem.tsukuba.ac.jp (K.M.); sugamata@chem.tsukuba.ac.jp (K.S.); 4Tsukuba Research Center for Energy Materials Sciences (TREMS), University of Tsukuba, 1-1-1 Tennoudai, Tsukuba 305-8571, Ibaraki, Japan

**Keywords:** disila[2]ferrocenophane, 9,9′-bi-9*H*-9-silafluorene, silyl anion, disilane, redox behavior

## Abstract

A disila[2]ferrocenophane bearing a 9,9′-bi-9*H*-9-silafluorene (9-silafluorene dimer) moiety as a bridging unit was synthesized and isolated as a stable crystalline compound. Disila[2]ferrocenophane **1,** newly obtained in this study, has been characterized by spectroscopic analyses, single crystal X-ray diffraction (SC-XRD) analysis, and electrochemical measurements. It was found that the obtained disila[2]ferrocenophane was reduced by a reducing agent to generate the corresponding 1,1′-ferrocenediyl-bis(silylanion) via the reductive Si–Si σ-bond cleavage. The trapping reactions of the 1,1′-ferrocenediyl-bis(silylanion) thus generated with electrophiles have also been attempted.

## 1. Introduction

The strategic integration of transition metals into supramolecular architectures presents a compelling avenue for the development of materials exhibiting unprecedented physical properties. This approach leverages the inherent capacity of transition metals to engender a spectrum of intriguing phenomena, including cooperative interactions, thereby unlocking functionalities unattainable through conventional synthetic routes [1,2]. Notably, the incorporation of transition metals within oligosilane frameworks has emerged as a particularly effective paradigm for the creation of materials endowed with novel optical, electronic, and redox characteristics [3,4]. Among such systems, oligosila[n]ferrocenophanes, where ferrocene units are integrated within oligosilane-based backbones, have garnered significant attention due to their unique structural features and reactivity [5,6,7,8,9]. Sila[1]ferrocenophanes, the simplest members of this class, have been extensively investigated, particularly in the context of their ring-opening polymerization (ROP), leading to the formation of –(fc-SiR_2_)_x_– polymers (fc = 1,1′-ferrocenylidene, R = organic substituent) [8,9,10,11,12,13,14,15,16,17]. This facile ROP behavior is attributed to the inherent strain within the bridged ferrocenylidene moiety of sila[1]ferrocenophanes.

In contrast, disila[2]ferrocenophanes, where the two cyclopentadienyl (Cp) groups of the ferrocenyl moiety are bridged by a disilane unit (–SiR_2_–SiR_2_–), have been reported less frequently [8,9,10,11]. Due to the long Si–Si bond length, the two Cp rings of the ferrocenyl unit in disila[2]ferrocenophanes exhibit a more parallel orientation, resulting in reduced ring strain compared to sila[1]ferrocenophanes. This reduced strain manifests in enhanced thermal and photochemical stability. For instance, fc[–Si(CH_3_)_2_–Si(CH_3_)_2_–] (1,1′-ferrocenediyl-1,1,2,2-tetramethyldisilane), disila[2]ferrocenophane **A,** shown in Figure 1, remains stable below 350 °C, while sila[1]ferrocenophanes generally undergo ROP at significantly lower temperatures (e.g., 130 °C for fc[–Si(CH_3_)_2_–]). While less prone to ROP, the disilane bridge in disila[2]ferrocenophanes offers unique reactivity due to the intrinsic nature of the Si–Si σ-bond [18], including photolytic bond cleavage and transition metal-catalyzed addition reactions to CºC π-bonds, influenced by the electronic communication between the ferrocenyl centers and the disilane moiety [9,19].

Studies have shown that these structural and electronic features of disila[2]ferrocenophanes lead to distinctive redox behavior, providing valuable insights into the role of silicon in mediating electronic interactions within organometallic systems. Recognizing the potential of disila[2]ferrocenophanes to expand the frontiers of silicon-based organometallic chemistry [8,9,10,11], we sought to investigate the interplay between redox activity and structural stability in these systems. Given the electron-accepting properties of the 9-silafluorenyl unit due to the aromaticity of the corresponding 9-silafluorenyl anion [20,21,22,23], we hypothesized that incorporating this moiety into the disila[2]ferrocenophane framework would yield intriguing electronic properties [24], e.g., the reductive Si–Si bond dissociation [25]. In this study, we designed and synthesized a novel disila[2]ferrocenophane derivative featuring a 9,9′-bi-9*H*-silafluorene bridging unit, with the aim of exploring its structural features and redox behavior.

## 2. Results and Discussion

9,9-Dichloro-9-silafluorene (**2**) was prepared by treating silicon tetrachloride (SiCl_4_) with 1,1-dilithiobiphenyl [24], generated in situ from 1,1′-dibromobiphenyl and *n*-butyllithium (*n*-BuLi). The subsequent treatment of compound **2** with diethylamine (Et_2_NH) and triethylamine (Et_3_N) yielded aminochlorosilane **3** in a 95% yield (Figure 1). Given its sensitivity towards air/moisture under ambient conditions, it was directly used in the next step after filtration through Celite to remove inorganic salts without further purification. Aminochlorosilane **3** was then reacted with 1,1′-dilithioferrocene, complexed with two equivalents of tetramethylethylenediamine (tmeda) (Li_2_(tmeda)_2_fc) [26], to afford 1,1′-bis(9-amino-9-silafluorenyl)ferrocene (**4**) in ca. 90% yield, as judged by ^1^H NMR spectroscopy. Compound **4** was further converted to 1,1′-bis(9-chloro-9-silafluorenyl)ferrocene (**5**) in an 80% yield by treatment with acetyl chloride.

The subsequent reduction of **5** with KC_8_ in toluene at 70 °C resulted in the successful formation of disila[2]ferrocenophane **1**, which could be isolated in a pure form by recrystallization from toluene (65% yield). Notably, **1** exhibits remarkable thermal stability, showing a melting point at 93–94 °C in air, without any decomposition. The corresponding sila[1]ferrocenophane containing a 9-silafluorenylidene moiety (sila[1]ferrocenophane **E**) also exhibits thermal stability, decomposing into oligomeric products at a much higher temperature of 190 °C [17].

The molecular structure of **1**, featuring a 9,9′-bi-9*H*-silafluorene bridging moiety on the 1,1′-ferrocenylidene skeleton, was elucidated through single-crystal X-ray diffraction (SC-XRD) analysis (Figure 2). The crystal structure revealed a solvated toluene molecule within the unit cell, and it was also shown that the two 9-silafluorenylidene moieties adopt a V-shaped conformation with an angle of 79.8° between their planes. The packing structure of **1** shows a dimeric arrangement, likely stabilized by intermolecular π(biphenyl)-π(biphenyl) and CH-π(biphenyl) interactions [27,28,29,30,31,32] between the perpendicularly oriented 9-silafluorenylidene moieties (Figure 2b). The intermolecular π(biphenyl)-π(biphenyl) distance between these interacting moieties was determined to be ca. 3.64 Å. The Si–Si bond in **1** exhibits a near-perpendicular orientation with respect to the cyclopentadienyl (Cp) rings. This molecular arrangement indicates a pseudo-*C*_2v_ symmetry, characterized by a mirror plane that coincides with the C–Si–Si–C plane and another mirror plane that includes the Fe atom, oriented parallel to the two Cp planes. Notably, the two Cp planes are observed to be nearly superimposed when viewed from above. This is suggested from the dihedral angle (ϕ) between the Cp ring and the C–Si–Si–C planes, which is approximately 90° (Figure 2c). The Si–Si bond length of 2.351(2) Å falls within the typical range for Si–Si single bonds, suggesting a relatively strain-free disila[2]ferrocenophane skeleton compared to sila[1]ferrocenophane **E** bearing a bridging 9-silafluorenylidene moiety. An analysis of the geometric parameters α, β, and δ, as depicted in Figure 2c, which are commonly used to assess skeletal strain, further supports the notion of a slightly strained **1** (α = 2.8°, β = 11.9°, 12.3°). This reduced strain is not attributed to the 9-silafluorenyl unit but rather to the intrinsic structural characteristics of disila[2]ferrocenophanes. This conclusion should be supported by the observation of similar structural parameters (α = 11.1°, β = 4.2°, δ = 176.4°, ϕ = 86°) in disila[2]ferrocenophane **A** [9,10], which features a tetramethyldisilane bridging unit.

It should be noted that the theoretically optimized structure of disila[2]ferrocenophane **1** (**1_opt_**) at the B3PW91-D3(BJ)/6-311G(3d) level [33] is different from the experimental structure determined by SC-XRD (Figure 3). In the optimized structure **1_opt_**, the two cyclopentadienyl (Cp) rings do not exhibit the V-shape geometry but the nearly parallel geometry (α = 1.47°), while the silicon (Si) atoms are slightly displaced from the Cp planes (β = 18.7°), resulting in a slightly shorter Si–Si bond length of 2.309 Å. This optimized geometry also revealed a non-perpendicular orientation of the Si–Si bond axis with respect to the Cp rings (ϕ = 76.7°). Furthermore, the two 9-silafluorenyl rings displayed a parallel arrangement with an interplanar angle of approximately 17°. Particularly, the Atoms-in-Molecules (AIM) calculations [34,35] clearly indicated the presence of intramolecular π-π interactions between the two 9-silafluorenylidene units, characterized by several bond critical points (BCPs) and ring critical points (RCPs) (Figure 3a). However, these intramolecular π-π interactions in **1_opt_** are likely to induce significant skeletal strain due to the large β angles. The magnitude of this strain was estimated to be 18.8 kcal/mol, based on an idealized isodesmic reaction involving **1_opt_** and hexamethyldisilane to give 1,1′-bis(trimethylsilyl-9-silafluorenyl)ferrocene, as depicted in Figure 2.

To reconcile the discrepancy between the experimental and theoretical structures, the dimeric structure of **1** was subjected to theoretical structural optimization at the same level. The resulting optimized dimeric structure (**1_2opt_**) closely resembled the experimentally observed structure from the SC-XRD analysis. The AIM calculations [34,35] for **1_2opt_** confirmed the presence of intermolecular π-π interactions between the parallelly oriented 9-silafluorenylidene moieties, with an intermolecular distance of 3.22 Å, as evidenced by the presence of BCPs/RCPs between the two 9-silafluorenylidene units (Figure 3b). These intermolecular π-π interactions appear to stabilize the pseudo-*C*_m_ symmetric structure of **1**, leading to the experimentally observed dimeric arrangement in the crystalline state. The stabilization energy associated with the dimerization of **1_opt_** to form **1_2opt_** was computationally estimated to be 14.1 kcal/mol per molecule (Figure 2). By combining the results of these theoretical calculations, the strain energy of an isolated molecule of **1** was estimated to be ca. 4.7 kcal/mol.

The electrochemical properties of **1** were investigated using cyclic voltammetry (CV) and differential pulse voltammetry (DPV) in dichloromethane (CH_2_Cl_2_) at room temperature (Figure 4). The CV of **1** exhibited a pseudo-reversible one-electron redox couple in the oxidation region, with a half-wave potential (*E*_1/2_) of 0.19 V versus ferrocenium/ferrocene (FcH^+^/FcH). This oxidation potential is significantly lower than that observed for sila[1]ferrocenophane **E,** bearing a 9-silafluorenylidene bridging moiety (*E*_1/2_ = 0.34 V vs FcH^+^/FcH) [17]. In the theoretical calculations, while the highest occupied molecular orbital (HOMO) of sila[1]ferrocenophane **E** (−5.75 eV) is predominantly localized on the ferrocenyl moiety, the HOMOs of the optimized monomeric (**1_opt_**, −5.65 eV) and the dimeric structures of **1** (**1_2opt_**, −5.60 eV) exhibit a significant contribution from orbitals derived from both the ferrocenyl moiety and the Si–Si σ-bond, as depicted in Figure 5. Such orbital compositions likely contribute to the lower oxidation potential observed for **1**. In the reduction region, both CV and DPV measurements conducted in tetrahydrofuran (THF) and CH_2_Cl_2_ revealed no distinct peaks but rather gradual current waves, suggesting a continuous electron transfer process followed by the subsequent decomposition reactions upon reduction. The lowest unoccupied molecular orbitals (LUMOs) of **1_opt_** (−1.43 eV) and **1_2opt_** (−1.50 eV) are comparable to or slightly lower in energy than that of sila[1]ferrocenophane **E** (−1.43 eV), as their LUMOs are primarily localized on the 9-silafluorenyl moieties.

As anticipated from the frontier orbital analysis of **1_opt_**, its UV–Vis spectrum in THF solution displayed several absorptions around 300 nm, along with a very weak absorption at a maximum wavelength (λ_max_) of 440 nm (ε = 110) in the longest wavelength region. This long-wavelength absorption is likely attributed to HOMO-LUMO transitions with a ligand-to-metal charge transfer (LMCT) character (Figure 6). Time-dependent density functional theory (TD-DFT) calculations [33] were employed to simulate the absorption spectra of **1_opt_** and **1_2opt_** (Figure 7). Although a definitive assignment of the observed absorptions is challenging due to the complexity of the spectra, the simulated spectrum for **1_opt_** (Figure 7a) exhibits better agreement with the experimental spectrum, particularly for the longer wavelength absorptions around 460 and 510 nm and several absorptions around 310 nm, compared to the simulated spectrum for **1_2opt_** (Figure 7b). Based on these findings, it is currently postulated that **1** predominantly exists as the monomeric structure (**1_opt_**) rather than the dimeric structure (**1_2opt_**) in solution.

Theoretical considerations suggest that the chemical reduction of **1** could induce Si–Si bond cleavage. This should be supported by the significant contribution of the Si–Si σ* antibonding orbital to the LUMO of **1**, which should be further lowered by the hyperconjugation with the low-lying π* orbitals of the 9-silafluorenyl moieties. Indeed, theoretical calculations at the B3PW91-D3(BJ)/6-311G(3d) level predicted a positive adiabatic electron affinity of +0.64 eV for **1_opt_**, in contrast to the negative value of –0.10 eV calculated for disila[2]ferrocenophane **A**, which features a tetramethyldisilane bridging unit. These theoretical findings prompted us to conduct an experimental investigation into the chemical reduction of **1**. The treatment of **1** with potassium graphite (KC_8_) in the presence of 18-crown-6 in benzene at room temperature generated the corresponding bis(silylanion) **7**^2–^ in situ, resulting in a deep-brown solution [8]. The formation of **7**^2–^ was confirmed by trapping experiments. The reaction of **7**^2–^ with chlorotrimethylsilane (ClSiMe_3_) and butyl chloride (BuCl) yielded compounds **6** (61% yield) and **8a** (65% yield), respectively (Figure 3).

Although attempts were made to react **7**^2–^ with 1,4-dichlorobutane to form the corresponding disila[6]ferrocenophane or a tetramethylene-linked ferrocene-9-silafluorenylidene polymer, the only isolated product was the chlorobutyl-substituted 1,1′-bissilylferrocene **8b** (62% yield). Similarly, the reaction of **7**^2–^ with dimethyldichlorosilane afforded trisila[3]ferrocenophane **9** (37% yield) as the primary product, while the formation of the corresponding polymeric product was not observed under these conditions. Furthermore, heating a mixture of **7**^2−^and dichloro-9-silafluorene **2** at 60 °C yielded trisila[3]ferrocenophane **10**, incorporating a bridging 9-silafluorenylidene trimer unit. These results demonstrate the potential of **7**^2−^ as a versatile building block for the synthesis of [n]ferrocenophanes featuring 9-silafluorenylidene bridging moieties. The obtained trisila[3]ferrocenophanes **9** and **10** exhibited characteristic weak absorptions in their UV–Vis spectra at λ_max_ = 455 nm (ε = 190) and 451 nm (ε = 210), respectively. These absorptions are likely the characteristic features of 9-silafluorenylidene-bridged oligosila[n]ferrocenophanes.

The molecular structures of the trapped products **6**, **9**, and **10** were elucidated through SC-XRD analyses (Figure 8). In compound **6**, the two silyl groups at the 1 and 1′ positions of the central ferrocenyl moiety exhibit an anti-periplanar orientation, similar to the arrangement observed in bis(chlorosilane) **5** (Figure 8a,b). In both **5** and **6**, the 9-fluorenyl π-systems are oriented towards the central Fe atom, probably due to the orbital interaction between d(Fe) and π*(9-silafluorenyl). Notably, trisila[3]ferrocenophanes **9** and **10** exhibit minimal strain within their ferrocenyl skeletons. This is evidenced by the near-perpendicular orientation of the two Cp rings and the near-coplanar arrangement of the Si atoms at the 1 and 1′ positions with the respective *ipso*-Cp rings. The plane defined by the three silicon atoms (Si_3_ plane) is tilted approximately 60° away from the Cp rings.

## 3. Materials and Methods

### 3.1. General Information

All manipulations were performed under an argon atmosphere using standard Schlenk techniques. Solvents were purified according to established procedures, and residual water and oxygen were rigorously removed by bulb-to-bulb distillation from a potassium mirror prior to use. ^1^H, ^13^C, and ^29^Si NMR spectra were acquired on a Bruker (Mannheim, Germany) AVANCE-400 spectrometer (^1^H: 400 MHz, ^13^C: 101 MHz, ^29^Si: 79.5 MHz). The internal standard for ^1^H NMR spectra was residual C_6_D_5_H (7.16 ppm) in C_6_D_6_, while C_6_D_6_ (128.0 ppm) served as the internal standard for ^13^C NMR spectra. External SiMe_4_ (0.0 ppm) was used as the standard for ^29^Si NMR spectra. Multiplicities in ^13^C NMR spectra were determined using the DEPT technique. High-resolution mass spectra (HRMS) were obtained on either a JEOL (Tokyo, Japan) JMS-T100LP (DART) or a Bruker APCI-Q-TOF-MS compact mass spectrometer. Melting points were determined using a Büchi (Flawil, Switzerland) melting point apparatus M-565 and are reported uncorrected. 9,9-dichloro-9-silafluorene (2) was synthesized according to a previously published method [17].

### 3.2. Synthesis of 9-Diethylamino-9-Chloro-9-Silafluorene (**3**)

To a mixture of 9,9-dichlorosilafluorene **2** (10.9 g, 43.6 mmol), THF (60 mL), and triethylamine (6.0 mL, 43.6 mmol), 80 mL of a THF solution of diethylamine (4.6 mL, 43.6 mmol) was slowly added at 0 °C. The reaction mixture was stirred for 15 h at room temperature. The solvent was then removed under reduced pressure. Toluene was added to the residue, and the mixture was filtered through Celite^®^. The filtrate was evaporated under reduced pressure to afford product **3** as a pale-yellow solid (11.5 g, 43.6 mmol, 91%). **3**: a pale-yellow solid; m.p. 68–70 °C; ^1^H NMR (400 MHz, CDCl_3_) δ 7.79 (ddd, *J* = 7.8, 0.7, 0.7 Hz, 2H), 7.67 (ddd, *J* = 7.2, 1.3, 0.7 Hz, 2H), 7.47 (ddd, *J* = 7.6, 7.6, 1.4 Hz, 2H), 7.31 (ddd, *J* = 7.3, 7.3, 1.0 Hz, 2H), 3.08 (q, *J* = 7.1 Hz, 4H), 1.09 (t, *J* = 7.0 Hz, 6H). ^13^C{^1^H} NMR (101 MHz, CDCl_3_) δ 146.2(C), 133.3(C), 132.6(CH), 131.7(CH), 128.4(CH), 121.0(CH), 40.3(CH2), 15.8(CH_3_); ^29^Si{^1^H} NMR (79.5 Hz, C_6_D_6_) δ –6.87. HRMS (APCI), *m*/*z*: found: 288.0951 ([M+H]^+^), calcd. for C_16_H_19_NSiCl ([M+H]^+^): 288.0970.

### 3.3. Synthesis of 1,1′-bis(9-Amino-9-Silafluorenyl)ferrocene (**4**)

To a hexane solution of ferrocene (3.54 g, 19.1 mmol) in hexane (30 mL), we added TMEDA (6.2 mL, 42.0 mmol) and *n*-BuLi (15.0 mL, 2.69 M in hexane, 40.1 mmol) at room temperature and stirred for 20 h. The precipitated orange solid was washed three times with hexane, and a THF solution (50 mL) of **3** (11.0 g, 38.2 mmol) was added at –78 °C and stirred at room temperature for 15 h. All volatiles were removed under reduced pressure, dichloromethane was then added to the residue and filtered through Celite^®^, and the filtrate was removed to afford **4** as an orange solid (11.8 g, 17.2 mmol, 90%). **4**: an orange solid; m.p. 112.2–114.0 °C; ^1^H NMR (400 MHz, C_6_D_6_) δ 7.76 (d, *J* = 7.3 Hz, 4H), 7.72–7.68 (m, 4H), 7.32 (ddd, *J* = 7.5, 1.6, 1.6 Hz, 4H), 7.28 (ddd, *J* = 7.3, 1.2, 1.2 Hz, 4H), 3.83 (dd, *J* = 1.7, 1.7 Hz, 4H), 3.50 (t, *J* = 1.7, 1.7 Hz, 4H), 2.65 (q, *J* = 7.1 Hz, 8H), 0.78 (t, *J* = 6.9 Hz, 12H). ^13^C NMR (101 MHz, C_6_D_6_) δ 147.9(C), 138.0 (C), 133.4(CH), 130.8 (CH), 128.0z (CH), 121.1 (CH), 75.0 (CH), 73.1 (CH), 64.6 (C), 40.3 (CH_2_), 16.3 (CH_3_); ^29^Si NMR (79.5 Hz, C_6_D_6_) δ –7.45. HRMS (APCI), *m/z*: found: 688.2407 ([M]^+^), calcd. for C_42_H_44_N_2_Si_2_Fe ([M]^+^): 688.2388.

### 3.4. Synthesis of 1,1′-bis(9-Chloro-9-Silafluorenyl)ferrocene (**5**)

A solution of **4** (3.75 g, 5.44 mmol) and acetyl chloride (2.0 mL, 28.1 mmol) in benzene (100 mL) was stirred for 5 h. All volatiles were removed under reduced pressure and the residue was washed with diethyl ether and toluene to afford **5** as a yellow solid (2.69 g, 4.36 mmol, 80%). **5**: a yellow solid; m.p. 123.4–123.9 °C; ^1^H NMR (600 MHz, C_6_D_6_) δ 7.62 (dd, *J* = 7.2, 1.6 Hz, 4H), 7.60 (dd, *J* = 7.2, 1.6 Hz, 4H), 7.18 (ddd, *J* = 7.2, 1.6 Hz, 4H), 7.07 (ddd, *J* = 7.2, 1.6 Hz, 4H), 4.10 (dd, *J* = 3.2, 1.6 Hz, 4H), 4.02 (dd, *J* = 3.2, 1.6 Hz, 4H); ^1^H NMR (600 MHz, C_6_D_6_) δ 7.61 (d, *J* = 7.3 Hz, 4H), 7.52 (d, *J* = 7.7 Hz, 4H), 7.18 (ddd, *J* = 7.7, 7.7, 0.7 Hz, 4H), 7.07 (dd, *J* = 7.2 Hz, 4H), 4.10 (dd, *J* = 1.7, 1.7 Hz, 4H), 4.03 (dd, *J* = 1.6, 1.6 Hz, 4H). ^13^C NMR (151 MHz, C_6_D_6_) δ 147.4 (C), 134.3 (C), 133.4 (CH), 132.1 (CH), 128.6 (CH), 121.6 (CH), 74.7 (CH), 73.6 (CH), 65.0 (C); ^29^Si NMR (119 MHz, C_6_D_6_) No signal could be observed probably due to its low solubility. HRMS (APCI), *m*/*z*: found: 614.0168 ([M]^+^), calcd. for C_34_H_24_Si_2_Cl_2_Fe ([M]^+^): 614.0139.

### 3.5. Synthesis of Disila[2]ferrocenophane (**1**)

To a solution of **5** (264 mg, 0.429 mmol) in toluene (40 mL), we added KC_8_ (116 mg, 0.858 mmol) and stirred at 70 °C for 20 h. To the reaction mixture, we added additional KC_8_ (116 mg, 0.858 mmol) and stirred at 70 °C for 20 h. Additional KC_8_ (5.80 mg, 0.0429 mmol) was then added, and the mixture was stirred at 70 °C for 20 h. After filtration through Celite^®^, the filtrate was removed under reduced pressure, and recrystallization from toluene at a low temperature afforded **1** as an orange solid (152 mg, 0.280 mmol, 65%). **1**: an orange solid; m.p. 93.4–94.8 °C; ^1^H NMR (400 MHz, CDCl_3_) δ 8.04 (m, 4H), 7.76 (d, *J* = 7.7 Hz, 4H), 7.41 (ddd, *J* = 7.6, 1.4,1.4 Hz, 4H), 7.35 (ddd, *J* = 7.3, 1.1, 1.1 Hz, 4H), 5.02 (dd, *J* = 1.7, 1.7 Hz 4H), 4.48 (dd, *J* = 1.7, 1.7 Hz 4H). ^13^C{^1^H} NMR (101 MHz, C_6_D_6_) δ 148.1 (C), 136.4 (C), 134.2 (CH), 130.6 (CH), 128.0 (CH), 121.8 (CH), 75.4 (CH), 72.4 (CH), 67.6 (C); ^29^Si {^1^H} NMR (79.5 MHz, C_6_D_6_) δ –11.0. HRMS (DART), *m/z*: found: 544.0747 ([M]^+^), calcd. zzfor C_34_H_24_Si_2_Fe ([M]^+^): 544.0767. UV/vis (THF), 440 nm (ε = 110).

### 3.6. Synthesis of 1,1′-bis(9-Trimethylsilyl-9-Silafluorenyl)ferrocene (**6**)

A mixture of **1** (76 mg, 0.14 mmol), KC_8_ (38 mg, 0.28 mmol), 18-crown-6 (74 mg, 0.28 mmol), and benzene (10 mL) was stirred for 3 h at room temperature. To the reaction mixture, chlorotrimethylsilane (1.0 mL, 7.9 mmol) was added and stirred for 2 h at the same temperature. After filtration through Celite^®^, all the volatiles were removed under reduced pressure. The residue was purified by GPC (toluene) to afford **6** (59 mg, 0.086 mmol, 61%). **6**: an orange solid; m.p. 117.4–118.9 °C; ^1^H NMR (400 MHz, C_6_D_6_) δ 7.79 (d, *J* = 7.7 Hz, 4H), 7.70 (d, *J* = 7.4, 4H), 7.32 (ddd, *J* = 7.6, 1.4, 1.4 Hz, 4H), 7.24 (ddd, *J* = 7.6, 1.2, 1.2 Hz, 4H), 3.91 (dd, *J* = 1.8, 1.8 Hz, 4H), 3.78 (dd, *J* = 1.7, 1.7 Hz, 4H), 0.00 (s, 18H); ^13^C{^1^H} NMR (101 MHz, C_6_D_6_) δ 148.8 (C), 139.1 (C), 133.9 (CH), 130.1 (CH), 127.6 (CH), 121.7 (CH) 74.0 (CH), 72.1 (CH), 65.1 (C), −1.9 (CH_3_); ^29^Si{^1^H} NMR (79.5 MHz, C_6_D_6_) δ −18.0, −19.0. HRMS (APCI), *m*/*z*: found: 690.1722 ([m]+), calcd. for C_40_H_42_Si_4_Fe ([M]^+^): 690.1709.

### 3.7. Synthesis of 1,1′-bis(9-Butyl-9-Silafluorenyl)ferrocene (**8a**)

A mixture of **1** (30 mg, 0.055 mmol), KC_8_ (15 mg, 0.11 mmol), 18-crown-6 (29 mg, 0.11 mmol), and benzene (5 mL) was stirred for 3 h at room temperature. To the reaction mixture, 1-chlorobutane (1.0 mL, 9.6 mmol) was added and stirred for 2 h at the same temperature. After filtration through Celite^®^, all the volatiles were removed under reduced pressure. The residue was purified by GPC (toluene) to afford **8a** as an orange oil (24 mg, 0.036 mmol, 65%). **8a**, an orange oil; ^1^H NMR (400 MHz, C_6_D_6_) δ 7.75 (d, *J* = 7.8 Hz, 4H), 7.66 (ddd, *J* = 7.0, 1.3, 0.6 Hz, 4H), 7.31 (ddd, *J* = 7.6, 1.4, 1.4 Hz, 4H), 7.22 (ddd, *J* = 7.3, 1.1, 1.1 Hz, 4H), 3.95 (dd, *J* = 1.7, 1.7 Hz, 4H), 3.84 (dd, *J* = 1.8, 1.8 Hz, 4H), 1.39–1.27 (m, 4H), 1.16 (m, 4H), 1.02–0.92 (m, 4H), 0.69 (t, *J* = 7.3 Hz, 6H); ^13^C{^1^H} NMR (101 MHz, C_6_D_6_) δ 148.7(C), 138.0 (C), 133.7 (CH), 130.7 (CH), 127.8 (CH), 121.4 (CH) 74.3 (CH), 72.3 (CH), 65.5 (C), 26.6 (CH_2_), 26.6 (CH_2_), 14.5 (CH_2_), 13.8 (CH_3_); ^29^Si{^1^H} NMR (79.5 MHz, C_6_D_6_) δ −2.0. HRMS (APCI), *m/z*: Found: 658.21885 ([M]^+^), Calcd. for C_42_H_42_Si_2_Fe ([M]^+^): 658.21755.

### 3.8. Synthesis of 1,1′-bis(9-Chlorobutyl-9-Silafluorenyl)ferrocene (**8b**)

A mixture of **1** (28 mg, 0.051 mmol), KC_8_ (14 mg, 0.10 mmol), 18-crown-6 (27 mg, 0.10 mmol), benzene (5 mL) was stirred for 3 h at room temperature. To the reaction mixture, 1,4-dichlorobutane (1.0 mL, 8.9 mmol) was added and stirred for 2 h at the same temperature. After filtration through Celite^®^, all the volatiles were removed under reduced pressure. The residue was purified by GPC (toluene) to afford **8b** as an orange oil (23 mg, 0.032 mmol, 62%). **8b**: an orange oil; ^1^H NMR (400 MHz, C_6_D_6_) δ 7.74 (d, *J* = 7.8 Hz, 2H), 7.63 (d, *J* = 7.0 Hz, 2H), 7.31 (ddd, *J* = 7.6, 1.4, 1.4 Hz, 2H), 7.22 (ddd, *J* = 7.4, 1.0, 1.0 Hz, 2H), 3.92 (dd, *J* = 1.7 Hz, 2H), 3.84 (dd, *J* = 1.8 Hz, 2H), 2.93–2.86 (m, 2H), 1.35–1.23 (m, 4H), 0.81–0.73 (m, 2H); ^13^C{^1^H} NMR (101 MHz, C_6_D_6_) δ 148.7(C), 137.6 (C), 133.7 (CH), 130.8 (CH), 127.9 (CH), 121.5 (CH), 74.3 (CH), 72.4 (CH), 65.1 (C), 44.2 (CH_2_), 35.9 (CH_2_), 21.5 (CH_2_), 13.7 (CH_3_); ^29^Si{^1^H} NMR (79.5 MHz, C_6_D_6_) δ −2.3. HRMS (APCI), *m*/*z*: Found: 726.1407 ([M]^+^), Calcd. for C_42_H_40_Si_2_Cl_2_Fe ([M]^+^): 726.1391.

### 3.9. Synthesis of Trisila[3]ferrocenophane (**9**)

A mixture of **1** (100 mg, 0.184 mmol), KC_8_ (49.6 mg, 0.368 mmol), 18-crown-6 (97.1 mg, 0.368 mmol), and benzene (15 mL) was stirred for 3 h at room temperature. To the reaction mixture, a benzene solution of dichlorodimethylsilane (0.42M, 0.44 mL, 0.184 mmol) was added and stirred for 15 h at the same temperature. All the volatiles were removed under reduced pressure, and then, the residue was filtered through Celite^®^ with benzene. The filtrate was evaporated and purified by GPC (toluene) to afford **9** as an orange solid (41.2 mg, 0.068 mmol, 37%). **9**: orange solid; m.p. 180 °C (decomp.); ^1^H NMR (400 MHz, C_6_D_6_) δ 8.00–7.98 (m, 4H), 7.76–7.73 (m, 4H), 7.31–7.24 (m, 8H), 4.49 (dd, *J* = 1.8, 1.8 Hz, 4H), 4.19 (dd, *J* = 1.8, 1.8 Hz, 4H), 0.04 (s, 6H); ^13^C{^1^H} NMR (101 MHz, C_6_D_6_) δ 148.8 (C), 138.1 (C), 134.3 (CH), 130.3 (CH), 127.7 (CH), 121.9 (CH) 74.7 (CH), 72.2 (CH), 66.4 (C), −4.5 (CH_3_); ^29^Si{^1^H} NMR (79.5 MHz, C_6_D_6_) δ −19.5, −37.0. HRMS (APCI), *m*/*z*: Found: 602.1019 ([M]^+^), Calcd. for C_36_H_30_Si_3_Fe ([M]^+^): 602.1000. UV/vis (THF), 455 nm (ε = 190).

### 3.10. Synthesis of Trisila[3]ferrocenophane (**10**)

A mixture of **1** (136 mg, 0.250 mmol), KC_8_ (67.5 mg, 0.499 mmol), 18-crown-6 (132 mg, 0.499 mmol), and benzene (15 mL) was stirred for 3 h at room temperature. To the reaction mixture, 9,9′-dichloro-9-silafluorene (62.7 mg, 0.201 mmol) was added and stirred for 15 h at 60 °C. All the volatiles were removed under reduced pressure, and then, the residue was filtered through Celite^®^ with benzene. The filtrate was evaporated and purified by GPC (toluene) to afford **9** as an orange solid (21.7 mg, 0.030 mmol, 12%). **10**: an orange solid; m.p. 135 °C (decomp.); ^1^H NMR (400 MHz, CDCl_3_) δ 7.74–7.72 (m, 4H), 7.62 (d, *J* = 7.8 Hz, 4H), 7.58–7.56 (m, 2H), 7.51 (d, *J* = 7.3 Hz, 2H), 7.29 (ddd, *J* = 7.6, 1.5, 1.5 Hz, 4H), 7.19 (ddd, *J* = 7.3, 1.0, 1.0 Hz, 6H), 7.07 (ddd, *J* = 7.3, 1.0, 1.0 Hz, 2H), 4.76 (dd, *J* = 1.7, 1.7 Hz, 4H), 4.59 (dd, *J* = 1.7, 1.7 Hz, 4H); ^13^C{^1^H} NMR (101 MHz, CDCl_3_) δ 148.6 (C), 148.3 (C), 136.2 (C), 135.9 (C), 134.0 (CH), 133.9 (CH), 130.2 (CH), 129.2 (CH), 127.4 (CH), 126.9 (CH), 121.4 (CH), 121.4 (CH), 74.7 (CH), 72.1 (CH), 67.0 (C); ^29^Si{^1^H} NMR (79.5 MHz, CDCl_3_) δ −21.9, −31.7. HRMS (APCI), *m*/*z*: Found: 724.1174 ([M]^+^), Calcd. for C_46_H_32_Si_3_Fe ([M]^+^): 724.1157. UV/vis (THF), 451 nm (ε = 210).

### 3.11. Electrochemical Measurements

Cyclic and differential pulse voltammograms were recorded on an BAS Inc. (Tokyo, Japan) ALS 1140A potentiostat/galvanostat using Pt wire electrodes under an argon atmosphere in custom-tailored glassware. Voltammograms were recorded at room temperature on CH_2_Cl_2_ solutions ([analyte]: 2.0 mM; supporting electrolyte: 0.1 M [*n*Bu_4_][PF_6_]) using a variety of scan rates (0.01–0.20 V/s). Only the most conclusive results obtained at a scan rate of 0.05 V/s are shown in Figure 4.

### 3.12. Measurements of UV–Vis Spectra

UV–Vis spectra were obtained using a SHIMADZU (Tokyo, Japan) UV-3150 UV–Vis-NIR spectrometer, employing 1 cm quartz cells fitted with J-Young (Cupertino, CA, USA) stopcocks, in a rigorously controlled argon atmosphere.

### 3.13. Theoretical Calculations

Density functional theory (DFT) calculations, encompassing geometry optimizations and frequency analyses of **1_opt_**, **1_2opt_**, and **6** were conducted using the Gaussian 16 (Revision C.01) software package [36]. Geometry optimizations were executed at the B3PW91-D3(BJ)/6-311G(3d) level of theory. The energetic minima of the optimized structures were validated through frequency calculations. Atoms in Molecules (AIM) analyses were performed using MultiWFN (version 3.7) [33,34]. Computational resources were generously allocated by the Supercomputer Laboratory at the Institute for Chemical Research (Kyoto University) and the Research Center for Computational Science, Okazaki, Japan (projects: 24-IMS-C377/24-IMS-C397). The Cartesian coordinates of the optimized structures are provided in the Appendix A.

### 3.14. X-Ray Crystallographic Analysis of 1, 5, 6, 9, and 10

Single crystals of **1**, **5**, **6**, **9**, and **10** were obtained through recrystallization from toluene. Preliminary diffraction data were collected at the BL02B1 beamline of SPring-8 (proposal numbers: 2022A1200, 2022A1354, 2022A1584, 2022A1626, 2022A1705, 2022B0552, 2022B0589, 2022B1626, 2023A1539, 2023A1771, 2023A1785, 2023A1794, 2023A1859, 2023A1925, 2023B1675, 2023B1806, 2023B1878, 2024A1633, 2024A1699, 2024A1851, 2024A1857, and 2024B2033) using a DECTRIS (Baden, Switzerland) PILATUS3 X CdTe 1M detector and synchrotron radiation (λ = 0.4135 Å). The structures were solved with SHELXT-2018 and refined by full-matrix least-squares on *F*² using SHELXL-2018 [35]. All non-hydrogen atoms were refined anisotropically, while hydrogen atoms were placed geometrically and refined using a riding model. Supplementary crystallographic data were deposited at the Cambridge Crystallographic Data Centre (CCDC) under deposition numbers CCDC-2421009 (**1**), CCDC-2421010 (**5**), CCDC-2421011 (**6**), CCDC-2421012 (**9**), and CCDC-2421013 (**10**); these can be obtained free of charge via www.ccdc.cam.ac.uk/data_request.cif (accessed on 16 March 2025).

Crystallographic data for **[1·(toluene)]** (CCDC-2421009): C_41_H_32_FeSi_2_, FW 636.69, crystal size 0.12 × 0.08 × 0.05 mm^3^, temperature –170 °C, orthorhombic, space group *P*_bca_ (#61), *a* = 13.3789(2) Å, *b* = 23.5018(4) Å, *c* = 19.3501(3) Å, *V* = 6084.22(17) Å^3^, *Z* = 8, *D*_calcd_ = 1.390 g cm^−3^, *μ* = 0.605 mm^−1^, *θ*_max_ = 26.000°, 103,740 reflections measured, 5954 independent reflections (*R*_int_ = 0.0815), 397 parameters refined, GOF = 1.174, completeness = 99.7%, *R*_1_ [*I* > 2*σ*(*I*)] = 0.0849, w*R*_2_ (all data) = 0.2267, largest diff. peak and hole 1.729 and –0.817 e Å^−3^. Crystallographic data for **5** (CCDC-2421010): C_34_H_24_Cl_2_FeSi_2_, FW 615.46, crystal size 0.15 × 0.10 × 0.05 mm^3^, temperature –170 °C, monoclinic, space group *P*2_1_/c (#14), *a* = 12.6567(7) Å, *b* = 12.1837(7) Å, *c* = 9.3354(5) Å, *β* = 108.0870(10)°, *V* = 1368.43(13) Å^3^, *Z* = 2, *D*_calcd_ = 1.494 g cm^−3^, *μ* = 0.858 mm^−1^, *θ*_max_ = 27.948°, 21,447 reflections measured, 3278 independent reflections (*R*_int_ = 0.0197), 178 parameters refined, GOF = 1.028, completeness = 99.5%, *R*_1_ [*I* > 2*σ*(*I*)] = 0.0255, w*R*_2_ (all data) = 0.0699, largest diff. peak and hole 0.447 and –0.180 e Å^−3^. Crystallographic data for **6** (CCDC-2421011): C_40_H_42_FeSi_4_, FW 690.94, crystal size 0.15 × 0.10 × 0.05 mm^3^, temperature –170 °C, monoclinic, space group *P*2_1_/c (#14), *a* = 13.3509(3) Å, *b* = 9.3007(1) Å, *c* = 15.9478(3) Å, *β* = 111.569(2)°, *V* = 1835.41(6) Å^3^, *Z* = 2, *D*_calcd_ = 1.250 g cm^−3^, *μ* = 0.568 mm^−1^, *θ*_max_ = 29.103°, 31,940 reflections measured, 4763 independent reflections (*R*_int_ = 0.0153), 289 parameters refined, GOF = 1.062, completeness = 96.7%, *R*_1_ [*I* > 2*σ*(*I*)] = 0.0259, w*R*_2_ (all data) = 0.739, largest diff. peak and hole 0.385 and –0.235 e Å^−3^. Crystallographic data for **9** (CCDC-2421012): C_36_H_30_FeSi_3_, FW 602.72, crystal size 0.10 × 0.05 × 0.03 mm^3^, temperature –170 °C, monoclinic, space group *C*2/c (#15), *a* = 25.2313(6) Å, *b* = 11.8580(3) Å, *c* = 19.4327(4) Å, *β* = 94.155(2)°, *V* = 5798.8(2) Å^3^, *Z* = 8, *D*_calcd_ = 1.381 g cm^−3^, *μ* = 0.669 mm^−1^, *θ*_max_ = 29.146°, 50,979 reflections measured, 7507 independent reflections (*R*_int_ = 0.0622), 363 parameters refined, GOF = 1.274, completeness = 95.9%, *R*_1_ [*I* > 2*σ*(*I*)] = 0.0757, w*R*_2_ (all data) = 0.1454, largest diff. peak and hole 0.494 and –0.366 e Å^−3^. Crystallographic data for **10** (CCDC-2421013): C_46_H_32_FeSi_3_, FW 724.83, crystal size 0.07 × 0.03 × 0.02 mm^3^, temperature –170 °C, monoclinic, space group *P*2_1_/c (#14), *a* = 15.1473(3) Å, *b* = 12.8955(2) Å, *c* = 18.2492(4) Å, *β* = 104.057(2)°, *V* = 3457.91(12) Å^3^, *Z* = 4, *D*_calcd_ = 1.392 g cm^−3^, *μ* = 0.575 mm^−1^, *θ*_max_ = 29.212°, 65,944 reflections measured, 8969 independent reflections (*R*_int_ = 0.0552), 451 parameters refined, GOF = 1.057, completeness = 95.6%, *R*_1_ [*I* > 2*σ*(*I*)] = 0.0409, w*R*_2_ (all data) = 0.0908, largest diff. peak and hole 0.465 and –0.315 e Å^−3^.

## 4. Conclusions

Disila[2]ferrocenophane **1**, which has a 9,9′-bi-9*H*-silafluorene moiety as a bridging unit with a 9-silafluorenyl unit, has been successfully synthesized. The SC-XRD structural analysis showed that it has a less strained structure relative to the corresponding sila[1]ferrocenophane **E**. While it was found that **1** is air/moisture stable in both solid and solution states, the chemical reduction of **1** was found to afford the corresponding dianion species **7**^2−^, which was experimentally supported by the trapping reactions with electrophiles such as chlorosilanes and chloroalkanes. It was demonstrated that 1 should be a potentially suitable building block for the creation of new oligosila[n]ferrocenophanes and Si/Fe-containing supermolecules.

## Data Availability

The raw data supporting the conclusions of this article will be made available by the authors upon request.

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
