# Peer review of "A Disila[2]ferrocenophane with a Bridging 9,9′-Bi-9H-9-Silafluorene Moiety"

_molecules, 2025, doi:10.3390/molecules30061361_

Round 1

Reviewer 1 Report

Comments and Suggestions for Authors

I have carefully reviewed the manuscript entitled “A Disila[2]ferrocenophane with a Bridging 9,9′-Bi-9H-9-silafluorene Moiety”. This study presents the successful synthesis and characterization of a new disila[2]ferrocenophane featuring a unique bridging 9,9′-bi-9H-silafluorene moiety. The work is well-executed and technically sound, and the synthesis of these challenging molecules is an important contribution to the field of silicon-containing organometallic chemistry.

Partcularly important is the creation of the novel disila[2]ferrocenophane, which has been thoroughly characterized using spectroscopic techniques, single-crystal X-ray diffraction (SC-XRD), and electrochemical studies. The manuscript provides additional discussions on the electronic properties and redox behavior of the system. The synthetic methodology is well-described and the theoretical calculations (level of theory, etc) are appropriate to complement the experimental findings.

 I recommend acceptance after minor revision. My only concern is that the authors did not mention the details of trisila[3]ferrocenophane 10 in terms of its UV/Vis absorption properties. Therefore I suggest the authors explicitly state how compound 10 behaves in UV/Vis spectroscopy. The authors should briefly discuss its absorption spectrum and compare it with the other ferrocenophane derivatives if needed.

Reviewer 2 Report

Comments and Suggestions for Authors

This manuscript presents the synthesis, characterization and electrochemistry of a disila[2]ferrocenophane. The work seems to have been done to the highest standards and it certainly warrants publication. I have a few minor comments I feel the authors should address before publication.

The opening sentence of the introduction had me slightly concerned with regard to how well the paper would be written. Fortunately, that was the worst sentence in the paper. I do think it could be made much clearer.

The authors quickly introduce the compound of interest as 1 but then often refer to it by the full name or at least a significant portion of the full name. I think it would read much better if they just consistently used 1 and not disila[2]ferrocenophane 1. 

On page 3 line 98 the authors suggest that 1 has pseudo-Cm symmetry. It seems to me that the authors are talking more about the geometry of a single molecule and so I would think C2v would be more appropriate. They also seem to not discuss how eclipsed the C5 rings are. This seems like something the authors should address 

On page 3 line 85, the authors are discussing the melting point of 1 and comparing it to the behavior of which oligomerizes at higher temperatures. This is a little confusing as they say that 1 exhibits remarkable thermal stability while it seems that E only decomposes at approximately 100 oC higher temperature. Can 1 be heated to that temperature or higher without oligomerization? This just needs some clarification otherwise I do not think it is an appropriate comparison. 

The later sections, were hard to read. I think a lot of that had to do with the placement of Scheme 3.

The authors perform the chemical reduction of 1 but make no mention of examining the reductive electrochemistry using CV. As they did CV to look at the oxidation, it would seem unusual to not look at the reduction. Perhaps it was outside the solvent window. It would be nice to know either way. But it would also be interesting to see if the compound go undergo chemically reversible Si-Si bonding breaking and formation. 

Was the electrochemistry in Figure 4 repeated at higher scan rates? Did the wave become more reversible? It would be nice to know this.

Several times in the experimental section the purification states that the filtrate was 'pumped up' and I am not sure what that means in this context.

Reviewer 3 Report

Comments and Suggestions for Authors

In this manuscript, the authors described the synthesis of an air stable disila[2]ferrocenophane featuring a 9,9’-bi-9H-silafluorene bridging moiety and study its structural features using multinuclear NMR spectroscopy, single crystal X-ray crystallography and cyclic voltammetry for redox studies. Furthermore, the experimentally investigated properties (structural, optical) are further supported by theoretically optimized structures of both its monomeric and dimeric forms. Interestingly, theoretically suggested, chemically in situ generated bis(silylanion) by KC8 reduction, was trapped by various chlorosilanes and chloroalkanes and the resulting products are structurally characterized by x-ray crystallography. I recommend accepting this article in the Molecules journal.

Minor suggestion:

i) The authors mention sila[1]ferrocenophane E in various places of the manuscript (eg, lines 85,102,160...), but its structure is not provided. It would be beneficial to include this structure in Figure 1.

ii)If possible, including images of π(biphenyl)-π(biphenyl) and CH-π(biphenyl) interactions from the X-ray structure in the supporting information would enhance visualization and understanding.
